# Single-photon test of hyper-complex quantum theories using a metamaterial

Lorenzo M. Procopio[1,*], Lee A. Rozema[1,*], Zi Jing Wong[2,3,*], Deny R. Hamel[1,4], Kevin O'Brien[2,3], Xiang Zhang[2,3], Borivoje Dakić[1,5] & Philip Walther[1]

In standard quantum mechanics, complex numbers are used to describe the wavefunction. Although this has so far proven sufficient to predict experimental results, there is no theoretical reason to choose them over real numbers or generalizations of complex numbers, that is, hyper-complex numbers. Experiments performed to date have proven that real numbers are insufficient, but the need for hyper-complex numbers remains an open question. Here we experimentally probe hyper-complex quantum theories, studying one of their deviations from complex quantum theory: the non-commutativity of phases. We do so by passing single photons through a Sagnac interferometer containing both a metamaterial with a negative refractive index, and a positive phase shifter. To accomplish this we engineered a fishnet metamaterial to have a negative refractive index at 780 nm. We show that the metamaterial phase commutes with other phases with high precision, allowing us to place limits on a particular prediction of hyper-complex quantum theories.

[1] Vienna Center for Quantum Science and Technology, University of Vienna, Boltzmanngasse 5, Vienna A-1090, Austria. [2] Nanoscale Science and Engineering Center, University of California, Berkeley, California 94720, USA. [3] Materials Sciences Division, Lawrence Berkeley National Laboratory, Berkeley, California 94720, USA. [4] Département de physique et d'astronomie, Université de Moncton, Moncton, New Brunswick E1A 3E9, Canada. [5] Institute for Quantum Optics and Quantum Information, Austrian Academy of Sciences, Boltzmanngasse 3, Vienna A-1090, Austria. * These authors contributed equally to this work. Correspondence and requests for materials should be addressed to X.Z. (email: xiang@berkeley.edu) or to P.W. (email: philip.walther@univie.ac.at).

Quantum mechanics is an extremely well-established scientific theory. Although it has been contested since its inception, beginning with the famous 'Bohr–Einstein debates[1],' quantum mechanics has stood its ground against competing theories and experimental tests for almost 100 years. Especially in the past decades, quantum mechanics has been challenged by a variety of alternative theories, including various hidden-variable models[2–11], non-linear modifications of quantum dynamics[12–17], spontaneous localization models[18–23] and generalized probabilistic theories[24–30]. In generalized probabilistic theories, sometimes also called 'post-quantum theories', quantum mechanics is just one particular theory in a vast sea of possibilities. One important class within this sea are the so-called hyper-complex theories[26–31]. They differ from standard quantum theory in the nature of superposition coefficients (probability amplitudes). Whether nature prefers a quantum theory based on real, complex, quaternionic or general hyper-complex amplitudes is an experimental issue. Excitingly, some predictions of hyper-complex are experimentally testable, since basing the superposition principle on hyper-complex probability amplitudes leads to a version of quantum mechanics wherein simple phases are not guaranteed to commute[32,33]. Although this prediction has experimentally been studied in the past with massive, non-relativistic particles[33], in that regime the quaternionic amplitudes are known to be exponentially suppressed[27,34]. Thus it is difficult to place bounds on genuine post-quantum effects in these experiments. However, new theoretical calculations have shown that this result does not necessarily apply to relativistic particles, such as single photons[35]. Therefore, any discrepancies between standard complex quantum theory and its hyper-complex generalization may be experimentally accessible in the relativistic regime.

The superposition principle states that linear combinations of wavefunctions are also valid wavefunctions. In textbook quantum mechanics these weighting coefficients are complex numbers, but there is no immediate theoretical requirement for this restriction. For example, it was shown by Birkhoff and von Neumann in 1936 that a mathematically consistent quantum theory can be constructed using only real numbers[36], but such a theory cannot correctly predict the results of certain experiments. One well-known example of this failure is that complex numbers are required to model all physically-realizable two-level systems, such as the polarization state of a photon. So far 'complex quantum mechanics' (CQM) has proven necessary to describe most quantum phenomena, but it is not known if it will remain sufficient.

Just as one can use real numbers, one can use hyper-complex numbers—such as quaternions[37]—to construct a quantum theory[26,27]. A quaternion is a mathematical generalization of the complex number with three, rather than one, imaginary components. Quaternionic quantum mechanics (QQM) has attracted much attention, in part because it is a natural and elegant extension of standard quantum theory[26–30,32,33,38,39]. Unlike many other post-quantum theories, QQM does not necessarily modify the postulates of quantum mechanics[24,25,40–42]. However, QQM makes certain experimental predictions which are different from the predictions of complex quantum mechanics— just as the predictions of a real quantum theory disagree with those of a complex theory.

One disagreement between CQM and QQM is the (non) commutativity of phases. In CQM phases commute, since they are described by complex numbers. However, in QQM phases are generally described by quaternions, which do not necessarily commute; thus, in QQM phases will not necessarily commute. On the basis of this idea, in 1979 Asher Peres proposed several experimental tests to search for quaternions in quantum

mechanics[32]. Because of technological limitations at the time, only a single neutron experiment has tested his ideas[33]. This work found a null result, which may not be surprising as it was later shown that quaternionic effects are likely to decay exponentially for massive particles[27]. Thus that experiment did not actually probe a prediction of quaternionic quantum theories. However, there is strong theoretical evidence that quaternionic effects will persist for relativistic particles, such as single photons. In fact, we recently showed that for relativistic Klein–Gordon scattering quaternionic effects do indeed persist[35]. Inspired by this and Peres' proposal, here we present an experiment using relativistic particles, that allows us to precisely search for the phase non-commutativity predicted by relativistic QQM.

To carry out our search for a hyper-complex effect we combine photonic quantum technologies, which provide a proven platform for foundational tests[6–11,43–46], with metamaterials[47–50] engineered to obtain a negative refractive index at 780 nm. We apply two different phases to single photons in a Sagnac interferometer, and perform a high-precision measurement to study their commutativity. We induce the two phases by very different optical media to enhance any potential non-commutativity. One phase is a standard optical phase (induced with a liquid-crystal), and the other is a negative phase which is induced by an artificial nanostructured metamaterial. Note that the phase is negative, in the sense that the Poynting vector points opposite to the propagation vector[51], see Supplementary Notes 2 and 3 for more details. The combination of a broadband, negative-index metamaterial with single-photon technology at optical wavelengths is a technological achievement. In our experiment we find that the net phase when applying the two phases in either order (meta-material before liquid crystal or vice versa) is equivalent to within at least 0.03°, meaning that complex quantum mechanics suffices to describe our experiment. To the best of our knowledge, our work places the most precise bounds on the commutativity of phases within hyper-complex quantum theories to date.

## Results

**Experimental proposal.** Our experiment is based on a Sagnac interferometer containing different phases. As illustrated in Fig. 1a, a perfectly balanced Sagnac interferometer, with an even number of reflections, results in all of the photons exiting through the same port that they entered. This results in a 'bright port' and a 'dark port'. However, this assumes that the phases commute, as CQM dictates. To be more specific, let $A$ and $B$ be two phase operators $A = \alpha \mathcal{I}$, and $B = \beta \mathcal{I}$ (where $\mathcal{I}$ is the identity operator). In CQM $\alpha$ and $\beta$ are complex numbers, but in general they could be quaternions, or other hyper-complex numbers. Then the probability to detect a photon in the dark port, in an ideal interferometer with no experimental imperfections, depends on the commutator of $\alpha$ and $\beta$ as

$$P_{\mathrm{D}}^{\mathrm{ideal}} = \frac{|[\alpha, \beta]|^2}{4}. \tag{1}$$

In CQM, $\alpha = e^{i\phi_A}$ and $\beta = e^{i\phi_B}$ are complex numbers, where $\phi_A$ and $\phi_B$ are real numbers. In this case $P_{\mathrm{D}}^{\mathrm{ideal}} = 0$. On the other hand, in QQM $\alpha$ and $\beta$ are quaternions, which do not generally commute; hence, we expect that $P_{\mathrm{D}}^{\mathrm{ideal}}$ can deviate from 0. See the Methods section, equation (11) for more details.

In practice, photons can also leak into the dark port because of experimental imperfections. We can quantify the imperfections of the Sagnac interferometer by a visibility, defined as $\nu = (P_{\mathrm{B}} - P_{\mathrm{D}})/(P_{\mathrm{B}} + P_{\mathrm{D}})$, that is $< 1$. Here, $P_{\mathrm{D}}$ ($P_{\mathrm{B}}$) is the probability to detect a photon in the dark (bright) port. In the Methods section we show that, for such an imperfect Sagnac

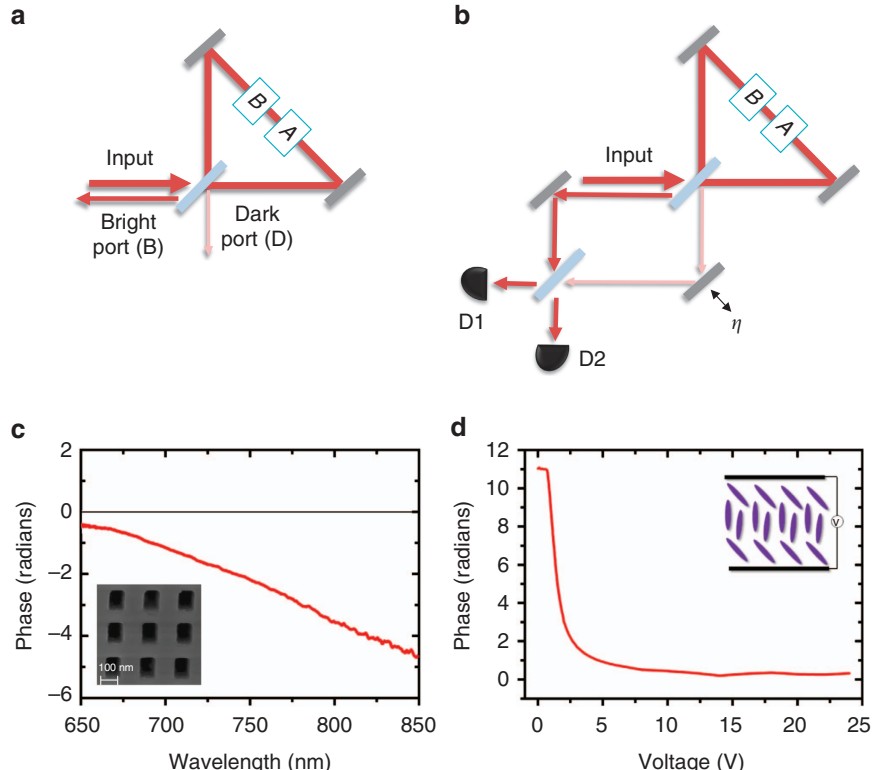

**Figure 1 | Experimental schematic and phase characterization.** (**a**) If two different phases A and B are placed inside a Sagnac interferometer and, if the phases commute, all the incoming light should exit through the 'bright port', while there should be no light in the 'dark port'. If A and B do not commute the dark port will not be dark. (**b**) Adding a Mach–Zehnder interferometer to interfere the bright and dark ports allows for a more precise measurement of the leakage into the dark port. (**c**) Wavelength dependence of the phase shift of our negative index metamaterial. For the wavelength of our single photons, 790 nm, the measured phase is about $-\pi$, which corresponds to a refractive index of the multilayer fishnet of $-0.4$. Inset: SEM image of the negative index metamaterial. (**d**) Phase response of the nematic liquid crystal. The measured relative phase (modulo $2\pi$) between the LC and the air for transmitted light is about $+\pi$. Inset: representation of a liquid crystal.

interferometer with two non-commuting phases, $P_D$ is

$$P_D = \frac{1}{2}\left(1 - v + \frac{v}{2}|[\alpha, \beta]|^2\right) \quad (2)$$

where $v$ is the visibility of Sagnac interferometer. Equation (2) is in fact a special case of the general equation presented in the Methods section (equation (14)), assuming that two phases commute with the reflection phase of the beamsplitter. Since we expect any deviation from CQM to be small, we expect $P_D$ to be small. Thus, we measure an amplified signal by interfering the bright and dark ports of the Sagnac interferometer in a Mach–Zehnder—like interferometer (Fig. 1b). (This visibility is amplified, with respect to a direct measurement of $P_D$, by a factor of $\sqrt{\frac{(1-P_D)}{P_D}}$). If the relative phase $\eta$ between these ports is scanned, the count rate in either output port of the Mach–Zehnder interferometer will oscillate as $P_{MZ} = \frac{1}{2} + \frac{1}{2}V\cos\eta$, where $V$ is the visibility of $P_{MZ}$:

$$V = \sqrt{1 - v^2\Gamma^2}. \quad (3)$$

where $\Gamma = 1 - \frac{|[\alpha,\beta]|^2}{2}$. Our goal is to measure this visibility $V$ experimentally when different phases are present in the Sagnac interferometer, and use this information to draw conclusions about the commutativity of the phases in our experiment via $\Gamma$. As we will see, if we perform two different measurements, each with different phases in the interferometer (two different values of $\Gamma$), we can altogether avoid needing to know $v$ the visibility of the Sagnac interferometer. To be clear, to use this fact

one must ensure that the visibility of the Sagnac interferometer is unchanged by the addition of the phase.

The choice of test phases is important for discovering potential quarternionic phases. In his proposal, Peres suggested an interferometry experiment using materials with complex scattering amplitudes, arguing that such materials would be more likely to have a quaternionic component. In this vein, we choose two optical materials with very different phase responses: one material with a positive refractive index, and one with a negative refractive index. We use a standard liquid-crystal phase retarder to provide a uniform, low-optical-loss phase shift as our first positive phase.

For our second phase we use an artificial nanostructured metamaterial. These materials have recently been used to probe several exciting quantum phenomena[52,53]. We designed our metamaterial to have a negative refractive index, and thus apply a negative phase. Achieving this requires both the real part of permittivity and permeability to be negative. We obtain this at optical frequencies with a fishnet optical metamaterial which integrates two types of structures together—one with a negative permittivity, and one with a negative permeability. See Supplementary Note 1 for more details.

**Interferometer performance.** A sketch of our experimental implementation is presented in Fig. 2. We send heralded single photons (see the Methods section) into a Sagnac interferometer. The Sagnac interferometer has two output modes, labelled B and D in Fig. 2. CQM predicts that B is the bright port, and D is the

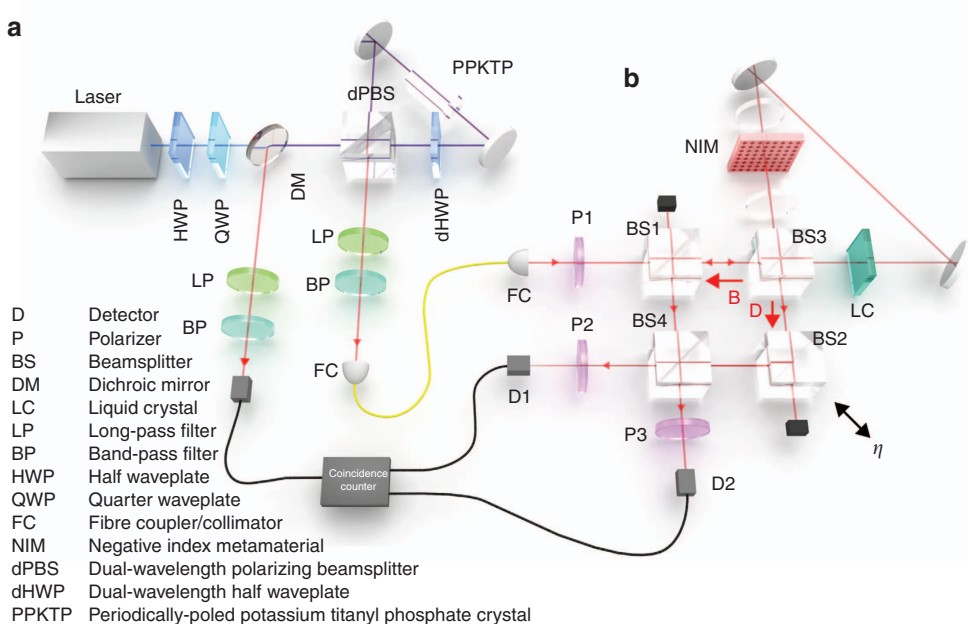

**Figure 2 | Experimental apparatus.** A detailed schematic of experiment to search for a quaternionic contribution to phase shifts. (**a**) We generate photon pairs in a separable polarization state. One photon is used to herald while the other one is sent to our interferometers. (**b**) We couple a Sagnac interferometer into a Mach–Zehnder interferometer to search for non-commuting phases. We monitor the interference in the Mach–Zehnder interferometer as its phase $\eta$ is scanned. The output photons are detected using single-photon detectors D1 and D2. The detectors are connected to coincidence logic to herald single photons. Two phases are applied inside the Sagnac which we can controllably 'turn on' and 'turn off'. The liquid crystal (LC) is controlled by applying voltage to it, and the negative index metamaterial (NIM) is mounted on a motorized translation stage so it can be 'turned off' by physically removing it from the interferometer.

dark port. After exiting the Sagnac interferometer, photons in mode B reflect off of BS1, and those in mode D reflect off of BS2. Beamsplitter BS2 is used to reflect mode D so that both modes experience the same attenuation, as this yields the highest visibility interference. The two modes then interfere at BS4. To ensure high-visibility interference, the input light is polarized with polarizer P1, and two final polarizers P2 and P3 (aligned to P1) are placed before the fibre couplers. Finally, both modes are coupled into single-mode fibre for spatial filtering.

To measure the interference between the B and D modes, BS2 is mounted on a piezo-actuated translation stage to scan the phase $\eta$. Because the visibility of the Sagnac interferometer is not perfect ($v<1$), we observe interference even without any phases in the Sagnac. This reference signal is shown in the bottom panel of Fig. 3a. In this plot, the counts registered at detector D1 are plotted versus the position of the translation stage. We also collect the photons exiting the other port, at detector D2 to normalize the data; these normalized data are plotted in the upper panel of Fig. 3a. We extract the visibility of the normalized curve by fitting the data, as described in the Methods section, and we find that the visibility is $V_o = 0.038 \pm 0.001$. The error is determined from the uncertainty of the fit parameters.

**Experimental characterization of the phases.** After characterizing our setup with no additional internal phases in the Sagnac interferometer we must characterize the individual effect of each of the two phases. We first turn on liquid-crystal phase retarder (LC) by applying a voltage that results in an effective phase of $\pi$ rad (see Fig. 1d for the details of our LC). A resulting interference signal, when $\eta$ the phase of the Mach-Zehnder interferometer is scanned, is shown in Fig. 3b. On a single run, turning the LC on does not introduce any measurable effects: the

visibility is still $V_{LC} = 0.038 \pm 0.001$. To reduce the influence of statistical fluctuations, this measurement is repeated 402 times. This minimizes the effects of long term noise, since each run is faster than any observable fluctuation. We find that the LC produces an average visibility difference of $\Delta_{LC} = V_{LC} - V_o = 0.002 \pm 0.003$. This result is consistent with 0, so we see turning on the LC has essentially no effect on our experimental apparatus. See Supplementary Fig. 8, for more details. This confirms that the visibility of the Sagnac interferometer $v$ is independent of the LC, and we can use this result to bound the systematic error induced by the LC. Note that these two measurements (presented in Fig. 3a,b) are only used to characterize our apparatus.

Next, we study the second phase: a negative phase shift of $-\pi$, which is induced by inserting the negative index metamaterial (NIM) into the Sagnac interferometer. The results of the negative-phase characterization are presented in Fig. 1c. Data with the NIM inserted and the LC phase set to 0 rad are shown in Fig. 3c. The NIM has a transmission of 13% at 790 nm, which is evident in the lower count rate of the raw data. We find that inserting the NIM marginally decreases the visibility of the Sagnac interferometer, leading to an increased Mach–Zehnder visibility of $V_{NIM} = 0.042 \pm 0.002$. This visibility increase occurs because inserting the NIM slightly degrades or shifts the spatial modes inside the Sagnac interferometer. We believe that this increase in visibility is rather a systematic error (that is, a decrease in the visibility of the Sagnac interferometer), and not the quaternionic effect that we are interested in.

**Data with both phases.** To observe an effect due to potential non-commutativity we only need study how visibilities change in response to different phases in the interferometer. Since the

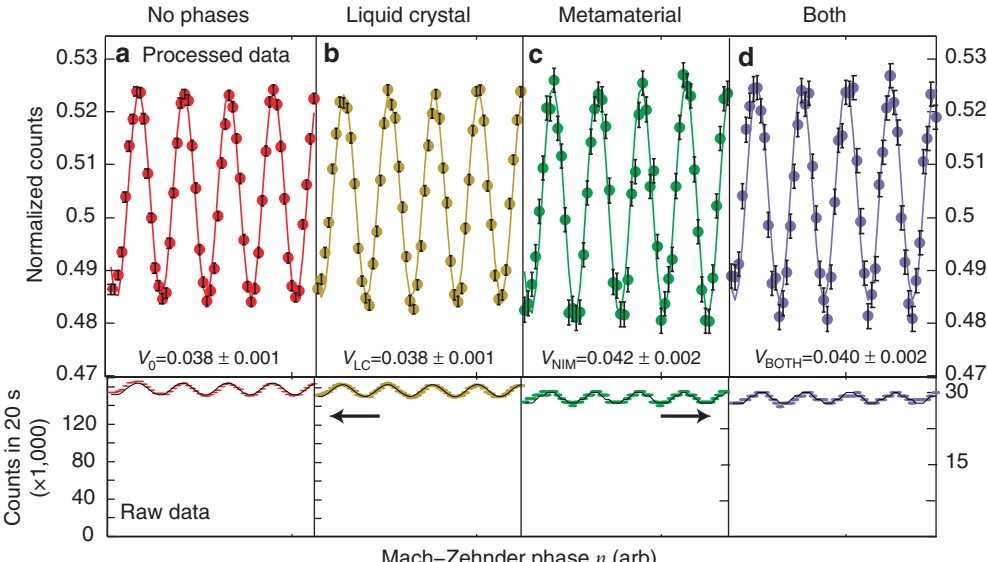

**Figure 3 | Representative Interferograms of the Mach–Zehnder Interferometer.** All of these data are photon counts exiting one port of the interferometer plotted versus the phase of the Mach–Zehnder interferometer. The data in the upper row are the count rates of photons exiting port one (at D1) normalized to the sum of the counts out our both ports. The lower row shows the same data without normalization. The error bars represent the standard error, arising from Poissonian counting statistics. Measurements for four cases are shown: (**a**) no phases inside of Sagnac loop, (**b**) only a positive phase (using the liquid crystal), (**c**) only a negative phase (using the negative-index metamaterial) and (**d**) both phases engaged. The visibilities of the data shown in **c,d** are equal within error. The unnormalized data (lower row) presented **a,b** uses the scale bar on the left, while the unnormalized data of **c,d** uses the scale bar on the right. The decreased count rate is due to the 13% transmission of the metamaterial.

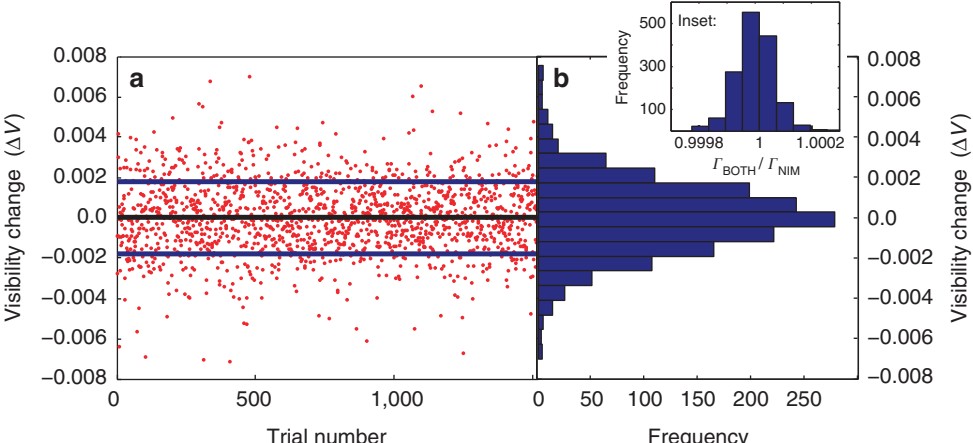

**Figure 4 | Results for repeated runs of the experiment.** (**a**) Each point corresponds to one run of the experiment, consisting of turning the liquid crystal off and measuring the visibility of the Mach–Zehnder interferometer, followed by turning the liquid crystal on and remeasuring the visibility. The difference between these two visibilities for each run is plotted here for data from each of the two ports of the Mach–Zehnder interferometer. A total of 761 experimental runs were made, resulting in 1,522 values of $\Delta V$. The black line marks the mean of all of the points, and the blue lines the standard deviation. (**b**) A histogram of data plotted in **a**. The error bars (from fitting to extract the visibility) on the individual points are not shown for clarity. Inset: A histogram of the values of $\frac{\Gamma_{\text{BOTH}}}{\Gamma_{\text{NIM}}}$, computed from the data in **a**. The mean value of this distribution is used to compute a phase difference between photons seeing the liquid–crystal phase retarder before or after the metamaterial. A mean value of this distribution that is $\neq 1$ would indicate some form of non-commutativity.

systematic error of the LC phase is much smaller than the error caused by inserting the NIM, we leave the NIM inserted and compare the visibility when the LC phase is set to 0 rad and $\pi$ rad. This allows us to neglect the larger systematic error of inserting the NIM.

Data with both the NIM inserted and LC phase set to $\pi$ are shown in Fig. 3d, and have a visibility of $V_{\text{BOTH}} = 0.040 \pm 0.002$. We need to compare this to the data presented in Fig. 3c.

On a single run the two visibilities are equal within experimental error, that is, $V_{\text{NIM}} = V_{\text{BOTH}}$. This already indicates that the two phases commute.

To decrease our statistical errors to the level of the LC systematic error, we repeat this experiment. We first set the LC to first to 0 rad and then $\pi$ rad a total of 761 times, while leaving the NIM inserted the entire time. In other words, we generate the data presented in Fig. 3c,d many times. For each run

we measure $\Delta V = V_{\text{BOTH}} - V_{\text{NIM}}$ for the data collected out of both ports at detectors D1 and D2, yielding a total of 1,522 values for $\Delta V$. These data are shown in Fig. 4a, and a histogram of these results is presented in Fig. 4b. Notice that on a given trial, $V_{\text{NIM}}$ can appear larger than $V_{\text{BOTH}}$, leading to a negative value. However, within error $V_{\text{NIM}}$ and $V_{\text{BOTH}}$ are equal for most trials. To be more precise, we examine the mean value of this distribution $\Delta V = 0.0006 \pm 0.005$. This is consistent with zero, and it indicates that the two phases in our experiment commute with a very high precision. The statistical error on $\Delta V$ is 0.005, which is slightly larger than the systematic error coming from turning on the LC.

## Discussion

As a final step we convert our visibility change into a different figure of merit to provide physical insight into our results: namely, a net phase difference when the NIM phase is applied before or after the LC phase. To start this conversion, we extract the ratio of $\Gamma$ when both phases are activated to $\Gamma$ when only the NIM is inside the Sagnac, $\Gamma_{\text{BOTH}}/\Gamma_{\text{NIM}}$, from the following definition

$$\frac{\Gamma_{\text{BOTH}}}{\Gamma_{\text{NIM}}} = \sqrt{\frac{1 - V_{\text{BOTH}}^2}{1 - V_{\text{NIM}}^2}}. \tag{4}$$

Here, $\Gamma_{\text{BOTH}}$ is defined in equation (17), and $\Gamma_{\text{NIM}}$ is defined in equation (19) of the Methods section. If this ratio deviates from 1, then there must be some non-commutativity. We can further convert this ratio into a phase shift between the clockwise and counter-clockwise modes of the the Sagnac interferometer simply as $\theta = \text{acos}\left(\frac{\Gamma_{\text{BOTH}}}{\Gamma_{\text{NIM}}}\right)$. See the Methods section for more details. We use equation (4) to compute $\Gamma_{\text{BOTH}}/\Gamma_{\text{NIM}}$ for every data point, the resulting distribution is shown in the inset of Fig. 4b. From the mean of this distribution we find $\Gamma_{\text{BOTH}}/\Gamma_{\text{NIM}} = 1$ with a precision of $2 \times 10^{-7}$, that is, $\Gamma_{\text{BOTH}}/\Gamma_{\text{NIM}} = 0.99999999 \pm 2 \times 10^{-7}$. Converting this a phase shift yields a bound of $\theta = 0.03°$.

In light of this analysis, our result can be seen as an extremely high-precision measurement of a phase shift between the two modes of the Sagnac interferometer. In principle, such a phase shift could arise from other effects, even in a common-path Sagnac interferometer such as ours. However, in our estimation, all of these potential phase shifts are orders of magnitude smaller than our null result. For example, given the geometry of our interferometer, the rotation of the Earth could lead to a phase shift of at most $10^{-4}$ degrees; Faraday effects caused by the Earth's magnetic field would be even smaller. Moreover, since they would be constant, all such phase shifts would present themselves as a reduced visibility of the Sagnac interferometer. In our experiment, the visibility of the Sagnac interferometer is not perfect; we attribute this to a slight mismatch between the spatial modes of the Sagnac interferometer. Given the polarizers before and after the interferometer, polarization mismatch between the two modes, although possible, is very small. We observed that any effect of the polarization mismatch is smaller than the spatial mismatch of the two modes. Again, these effects lead to a systematic decrease in the visibility of the Sagnac interferometer, and our data analysis accounts for this.

The phase shift derived here also allows us to compare our result to a previous neutron interferometry experiment[33]. Although we should point out that the deviation from CQM could be different for neutrons and photons, and thus such tests must be carried out in a variety of physical systems. In fact, quaternionic effects would likely decay exponentially for neutrons[27]. In the neutron experiment it was found that two interference patterns (each created with two phases

shifters inserted in either order) were shifted by less than 0.3°. Then, since each phase shifter imparted a phase on the order of 10,000°, they concluded that any quaternionic contribution must be <1 part in 30,000. However, this assumes that the quaterionic phase is linearly proportional to total phase—there is no such requirement in QQM (see the Methods section). In fact, the quaternionic phase could be completely independent of the standard quantum phase. Thus only the absolute deviation from CQM's predictions is relevant to the quaternionic non-commutativity, and relevant bound from the previous work is 0.3°—our bound is one order of magnitude tighter than this.

In our work we directly probe quaternionic quantum mechanics using relativistic particles[38]. A previous experiment used neutron interferometry, but it has been theoretically predicted that non-trivial quaternionic effects are exponentially suppressed for non-relativistic particles (such as neutrons). However, this has not been proven for relativistic particles, such as single photons. In fact, it was shown that quaternionic effects for relativistic Klein-Gordon scattering can persist[35]. This motivates our work, wherein we directly search for quaternionic effects within a relativistic framework. Our work was enabled by the combination of a novel negative-index metamaterial with standard optical photonic technology, but further tests of QQM could be performed within optics using other methods to apply phases, or in other regimes (using, for example, near-field measurements). Further tests with other massive particles (that is, using molecular, electron, or other matter-wave interferometers) could also prove fruitful, but therein the measurements must be made extremely carefully to probe for exponentially decaying effects[27]. Regardless of the physical system, it is essential to continue to search for effects predicted by post-quantum theories, as such tests may one day point towards a future theory, supplanting quantum mechanics.

## Methods

**Single-photon source.** Our single-photon source is based on a Sagnac interferometer, commonly used to create polarization-entangled photon pairs, but we generate photon pairs in a separable polarization state. Our Sagnac loop is built using a dual-wavelength polarizing beamsplitter (dPBS) and two mirrors. A type-II collinear periodically-poled Potassium Titanyl Phosphate (PPKTP) crystal of length 20 mm is placed inside the loop and pumped by a 23.7 mW diode laser centred at 395 nm. This results in photon pairs at a degenerate wavelength of 790 nm. The pump beam polarization is set to horizontal in order to generate the down-converted photons in a separable polarization state $|H\rangle|V\rangle$. The dichroic mirror (DM) transmits the pump beam and reflects the down-converted photons, and the half wave plate (HWP) and quarter waveplate (QWP) are used to adjust the polarization of the pump beam. Long (LP) and narrow band (BP) pass filters block the pump beam and select the desired down-converted wavelength. Polarizers are aligned to transmit only down-converted photons with the desired polarization. After this, the down-converted photon pairs are coupled into single-mode fibres (SMF), and one photon from the pair is used as a herald while the other single photon is sent to the rest of the experiment using a fibre collimator (FC).

**Theoretical treatment of the Sagnac interferometer.** Here we derive the probability of a photon incident on an imperfect Sagnac interferometer to exit the 'dark port' if two phases internal to the Sagnac interferometer do not commute.

We start with a single-photon incident on a 50:50 beamsplitter. Ideally, given a 50:50 beamsplitter and a reflection phase of $\pi/2$, the state of a photon after reflecting is:

$$\left(i|1,0\rangle_{\text{CW,CCW}} + |0,1\rangle_{\text{CW,CCW}}\right)/\sqrt{2}, \tag{5}$$

where CW and CCW refer to the clockwise and counter-clockwise modes in Fig. 1a, respectively. Next, applying two phases (as in Fig. 1a), represented by operators $A$ and $B$, we have

$$\left(ABi|1,0\rangle_{\text{CW,CCW}} + BA|0,1\rangle_{\text{CW,CCW}}\right)/\sqrt{2}. \tag{6}$$

To be completely general we will assume that the '$i$' does not commute with $A$ and $B$. The operators $A$ and $B$ can be represented as

$$A = \alpha\mathcal{I}, \quad B = \beta\mathcal{I}, \tag{7}$$

where $\mathcal{I}$ is the identity operator. In complex quantum mechanics $\alpha = e^{i\phi_A}$ and $\beta = e^{i\phi_B}$, where $\phi_A$ and $\phi_B$ are real numbers. In this case, $\alpha$ and $\beta$ are complex numbers so $A$ and $B$ commute. However, in quaternionic quantum mechanics the phase $\phi_A$ is generalized to vector $\{\phi_A^1, \phi_A^2, \phi_A^3\}$, where $\phi_A^1, \phi_A^2$, and $\phi_A^3$ are real numbers. Then $i\phi_A$ is replaced with $i\phi_A^1 + j\phi_A^2 + k\phi_A^3$, where $\{i, j, k\}$ is a basis over the imaginary part of the quaternionic space. With these definitions $\alpha$ and $\beta$ in equation (7) become unit quaternions

$$\alpha = e^{i\phi_A^1 + j\phi_A^2 + k\phi_A^3}, \qquad \beta = e^{i\phi_B^1 + j\phi_B^2 + k\phi_B^3} \qquad (8)$$

Now, the operators $A$ and $B$ of equation (7) no longer commute in general. In fact, $\alpha$ and $\beta$ could be even more general hyper-complex numbers, consisting of more than three imaginary components.

Next, by applying the form of the operators defined in equation (7), we can write the state in equation (6) as

$$\left( \alpha\beta i |1, 0\rangle_{\mathrm{CW,CCW}} + \beta\alpha |0, 1\rangle_{\mathrm{CW,CCW}} \right) / \sqrt{2}. \qquad (9)$$

In complex quantum mechanics, $\alpha\beta = \beta\alpha$ and the two complex numbers describe a global phase, so they have no effect on experimental outcomes. However, if $\alpha$ and $\beta$ do not commute, the output state is

$$\frac{1}{2} (\alpha\beta i + i\beta\alpha) |1, 0\rangle_{\mathrm{B,D}} + \frac{1}{2} (i\alpha\beta i + \beta\alpha) |0, 1\rangle_{\mathrm{B,D}}. \qquad (10)$$

Thus the probability for an incident photon to exit the Sagnac interferometer via the dark port (the amplitude of the second term) is

$$P_{\mathrm{D}}^{\mathrm{ideal}} = \frac{1}{4} |i\alpha\beta - \beta\alpha i|^2. \qquad (11)$$

This quantifies the degree of commutativity between $\alpha$, $\beta$, and $i$. If $\alpha$, $\beta$, and $i$ all mutually commute it is zero. Moreover, if $i$ commutes with $\alpha$ and $\beta$ it simply becomes the commutator of $\alpha$ and $\beta$, as shown in equation (1) of the main text.

We will next treat the imperfect alignment of our interferometer. We start by writing the state from equation (9) as a density matrix

$$\frac{1}{2} \begin{pmatrix} 1 & \alpha\beta i \alpha^* \beta^* \\ -\beta\alpha i \beta^* \alpha^* & 1 \end{pmatrix}, \qquad (12)$$

where the $*$ denotes the conjugate of a quaternion or complex number. Let our Sagnac interferometer have a visibility of $v = (P_B - P_D)/(P_B + P_D)$, where $P_D$ and $P_B$ are the intensities of the dark and bright ports, respectively. We can model this by simply scaling the coherences by $v$, as

$$\frac{1}{2} \begin{pmatrix} 1 & v\alpha\beta i \alpha^* \beta^* \\ -v\beta\alpha i \beta^* \alpha^* & 1 \end{pmatrix}. \qquad (13)$$

This reduced coherence can be derived by coupling the CW and CCW modes to additional modes, and then tracing out those additional modes. This is a very general method to model imperfections since it does not require any assumptions on the types of imperfections: the CW and CCW modes could couple to additional spatial modes, temporal modes, etc.

Again, we can compute the probability to find the photon in the dark port by applying the beamsplitter transformation. Doing so yields

$$P_{\mathrm{D}} = \frac{1}{2} \left( 1 - v + \frac{v}{2} |\alpha\beta - \beta\alpha i|^2 \right), \qquad (14)$$

Then the probability of the photon to exit the bright port is simply $P_B = 1 - P_D$. Notice that $P_D$ defined here differs slightly from the equation (2) of the main text, in that $|i\alpha\beta - \beta\alpha i|$ replaces $|[\alpha, \beta]|$. However, a non-zero value of this new quantity would also signify a deviation of QQM from CQM, and is, thus, also interesting to study.

**Theoretical treatment of the Mach-Zehnder interferometer.** After the Sagnac interferometer, the bright and dark ports are interfered in our Mach–Zehnder interferometer (Fig. 1b). Interfering two optical fields, with intensities of $P_B$ and $P_D$, on a 50:50 beamsplitter results in a signal with a visibility of.

$$V = 2\sqrt{P_B P_D}, \qquad (15)$$

The same result holds if $P_B$ and $P_D$ are instead the probabilities of finding a photon in either path. Thus, the visibility of the Mach–Zehnder interferometer with both phases inserted in the Sagnac interferometer, can be computed from $P_D$ (equation (14)). After simplifying, we arrive at

$$V_{\mathrm{BOTH}} = \sqrt{1 - v^2 \Gamma_{\mathrm{BOTH}}^2}, \qquad (16)$$

where

$$\Gamma_{\mathrm{BOTH}} = 1 - \frac{1}{2} |i\alpha\beta - \beta\alpha i|^2 \qquad (17)$$

This visibility $V_{\mathrm{BOTH}}$ is a function of both the degree of commutativity $|i\alpha\beta - \beta\alpha i|$ and the visibility of the Sagnac interferometer $v$. To compare to our experimental procedure imagine that we turn off the liquid–crystal phase (which we represent by $\alpha$) and leave the negative-index metamaterial inserted. Then $\alpha$ drops out and the degree of commutativity becomes the commutator of $i$ and $\beta$, so equation (16)

becomes

$$V_{\mathrm{NIM}} = \sqrt{1 - v^2 \Gamma_{\mathrm{NIM}}^2}, \qquad (18)$$

where

$$\Gamma_{\mathrm{NIM}} = 1 - \frac{1}{2} |[i, \beta]|^2. \qquad (19)$$

This visibility that depends on the commutation of the negative-index metamaterial with the reflection phase inside the Sagnac interferometer, and on the visibility $v$ of the Sagnac interferometer.

By combining equations (16 and 18) we arrive at a result which does not depends only on two measurable visibilities of the Mach-Zehnder interferometer, and not on the visibility $v$ of the Sagnac interferometer:

$$\frac{1 - \frac{1}{2} |i\alpha\beta - \beta\alpha i|^2}{1 - \frac{1}{2} |[i, \beta]|^2} = \sqrt{\frac{1 - V_{\mathrm{BOTH}}^2}{1 - V_{\mathrm{NIM}}^2}} \equiv \frac{\Gamma_{\mathrm{BOTH}}}{\Gamma_{\mathrm{NIM}}}, \qquad (20)$$

Thus we can experimentally determine the ratio $\Gamma_{\mathrm{BOTH}}/\Gamma_{\mathrm{NIM}}$ from two visibilities of the Mach–Zehnder interferometer with and without the liquid–crystal phase turned on. The left-hand side of equation (20) simplifies to the $\Gamma$ defined after equation (3) in the main text if $i$ commutes with both $\alpha$ and $\beta$. Notice also that if $|i\alpha\beta - \beta\alpha i| = |[i, \beta]| \neq 0$ this ratio will be one. Thus this parameter is insensitive to a very specific type of non-commutativity between $\alpha$, $\beta$, and $i$ where-in $|i\alpha\beta - \beta\alpha i| = |i\beta - \beta i|$. Physically this would be the case, for example, if $\alpha$ commutes with $\beta$ and $i$, but $\beta$ and $i$ do not commute. The reason for defining the quantity $\Gamma_{\mathrm{BOTH}}/\Gamma_{\mathrm{NIM}}$ will become clear in the next section.

**Converting the visibility change into a phase change.** In this section we will derive a figure of merit which provides additional physical intuition into our results. Namely, a difference in the net phase between the NIM phase being applied before the LC phase, and vice versa. In our experiment we measure the visibility of an interference signal which is proportional to the commutator of the two phases. This signal arises from interference between the dark and bright output ports of the Sagnac interferometer. As we show above, if two phases inside the Sagnac do no commute, light will leak into the dark port. Then interfering the bright and dark modes leads to an interference signal which has a visibility given by equation (15).

Imagine that leakage into the dark port arises from a phase shift $\theta$ between the clockwise and the counter-clockwise modes of the Sagnac. Physically, this means that there is a different phase shift if the photon sees the metamaterial before or after the liquid-crystal. It is straightforward to show, within CQM, that if the two modes of a Sagnac interferometer experience a phase shift $\theta$ the probabilities of the photon exiting either port become

$$\begin{aligned} P_B &= \frac{1}{2} + \frac{v}{2} \cos\theta, \\ P_D &= \frac{1}{2} - \frac{v}{2} \cos\theta, \end{aligned} \qquad (21)$$

where $v$ is the visibility of the Sagnac interferometer. Now substituting equation (21) into equation (15) we arrive at the visibility of the Mach-Zehnder interferometer as a function of the phase inside the Sagnac interferometer

$$V(\theta) = 2\sqrt{1 - v^2 \cos^2\theta}. \qquad (22)$$

Experimentally, we measure two visibilities of the Mach–Zehnder interferometer, which we now attribute to a phase change in the Sagnac interferometer. In the present picture, $V(\theta)$ and $V(0)$ are the visibilities of the Mach-Zehnder interferometer with and without a phase difference between the clockwise and counter-clockwise modes. Thus, we will equate $V(0)$ to the visibility when only one phase is inside the Sagnac interferometer $V_{\mathrm{NIM}} \equiv V(0)$, and $V(\theta)$ to the visibility when both phases are in the Sagnac interferometer $V_{\mathrm{BOTH}} \equiv V(\theta)$. Then we will substitute equation (22) into (20), simplifying and solving for $\theta$. Doing this yields

$$\theta = \mathrm{acos} \left[ \sqrt{\frac{1 - V_{\mathrm{BOTH}}^2}{1 - V_{\mathrm{NIM}}^2}} \right] = \mathrm{acos} \left( \frac{\Gamma_{\mathrm{BOTH}}}{\Gamma_{\mathrm{NIM}}} \right). \qquad (23)$$

We can then understand this $\theta$ as an effective phase shift between the clockwise and counter-clockwise modes, arising from the non-commutativity of the phases. So we see that measuring these two visibilities allows us to use equation (23) to convert our result into this phase. Doing this, and using Gaussian error propagation on equation (23) results in $\theta < 0.03°$.

**Fitting to extract visibility.** To extract the visibility from the normalized data we fit a sinusoid to the data, and calculate the visibility from the fit parameters. The explicit form of our fitting equation is

$$\varepsilon \sin^2 (fx + p) + \kappa, \qquad (24)$$

where $\varepsilon$, $\kappa$, $f$, and $p$ are all free parameters. The visibility of this curve in

equation (24) in terms of the fit parameters is

$$\frac{\varepsilon}{\varepsilon + 2\kappa}. \tag{25}$$

We compute the error on each visibility using Gaussian error propagation, starting with the fitting uncertainties.

**Negative-index metamaterial.** We use a fishnet metamaterial to achieve an optical medium with a negative refractive-index. Our fishnet negative index metamaterial (NIM) consists of seven physical layers of silver (Ag, 40 nm) and magnesium fluoride (MgF$_2$, 50 nm), with a 15 nm capping layer of MgF$_2$. The metamaterial is suspended to avoid any positive phase contribution from the substrate. Figure 1c shows the resulting negative phase shift of our NIM as a function of the wavelength of the light, and the inset shows an SEM image of the surface of our NIM. Supplementary Notes 1–3 contains complete details of the design, fabrication, and characterization of our NIM.

In our experiment, the NIM is mounted on an automated translation stage so that it can be reliably and repeatably removed and inserted. It has a clear aperture of approximately 20 μm, thus we focus the beam sufficiently to pass through it. To find the optimal position of the NIM, we scan the translation stage, while monitoring the transmission of both the clockwise and counter-clockwise modes of the Sagnac interferometer. We align the sample, relative to the focus of the lenses, such that the transmission of both modes is maximized at the same position. Another point of concern is the significant back reflection ($\approx 50\%$) of the NIM for our wavelength range. Since this back reflection can couple to our detectors, we slightly tilt the NIM, by 0.44°, to reduce this background signal. We tilt the NIM along a carefully chosen axis so as to keep the polarization parallel to the thinner lines of the fishnet nanostructures, it has been shown that in this configuration such metamaterials still work optimally[54].

**Liquid crystal retarder.** We use a commercial nematic liquid crystal cell whose molecules orient to an applied electrical field. We characterize the LC by placing it between two polarizing beamsplitter cubes with its optical axis at an angle of 45°. We then measure the light intensity transmitted through the second PBS as we vary the voltage applied to the LC. Since the transmitted intensity is proportional to $\frac{1}{2}(1 + \cos \zeta)$, where $\zeta$ is the relative phase imparted by the LC, this measurement allows us to determine the relative phase (modulo $2\pi$) effected by the LC as a function of the applied voltage. The measured relative phase of the LC is shown in Fig. 1d.

**Data availability.** The data that support the findings of this study and the computer code to analyse it are available from the corresponding authors on request.

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

## Acknowledgements

We thank Stephen Adler, Časlav Brukner and Gregor Weihs for stimulating discussions, Michael Mrejen for his help in characterizing the metamaterial sample, and T. Rögelsperger for assisting with the figures. We acknowledge support from the European Commission, QUILMI (No. 295293), EQUAM (No. 323714), PICQUE (No. 608062), GRASP (No. 613024), QUCHIP (No. 641039) and RAQUEL (No. 323970); the Austrian Science Fund (FWF) through START (Y585-N20), the doctoral programme CoQuS (W1210-3), and Individual Project (No. 2462); the Vienna Science and Technology Fund (WWTF, grant ICT12-041); the United States Air Force Office of Scientific Research (FA9550-1-6-1-0004); L.M.P. acknowledges partial support from CONACYT-Mexico. L.A.R. was partially funded by a Natural Sciences and Engineering Research Council of Canada (NSERC) Postdoctoral Fellowship. Z.J.W., K.O. and X.Z. are supported by the U.S. Department of Energy, Office of Science, Basic Energy Sciences, Materials Sciences and Engineering Division under contract no. DE-AC02-05CH11231.

## Author contributions

L.M.P., B.D. and P.W. designed the experiment. L.M.P., D.R.H. and L.A.R. built the experiment. L.M.P. and L.A.R. collected the data and performed the final analysis. Z.J.W., K.O. and X.Z. designed, fabricated and characterized the metamaterial sample. B.D. provided the theoretical analysis and the concept for the experiment. B.D., X.Z. and P.W. supervised the project. All the authors contributed to the writing of the final manuscript.

## Additional information

**Competing interests:** The authors declare no competing financial interests.

**Publisher's note**: 

