## [Peer Review File · Nature Communications]

Reviewer #1

Referee report on NCOMMS-16-18605: Single-Photon Test of Hyper-Complex Quantum Theories

The manuscript presents an experimental test of a class of post-quantum theories, namely quaternionic quantum mechanics (QQM). A prediction of QQM is the non-commutativity of phases, in contrast to ordinary QM in which phases commute. The authors test experimentally this prediction for photons in a Sagnac interferometer with two different phases: a positive phase produced by a liquid crystal (LC) and a negative phase produced by a negative index metamaterial (NIM).

The article is well-written and the subject is of interest for quantum foundations. The results are timely and important for future developments of post-quantum theories, as they exclude a class of theories incompatible with experiments.

I have a few comments/questions for the Authors:

1. I suggest to rewrite eq.(2) as

$$P_D = \frac{1}{2} \left(1 - v + \frac{v}{2} |[\alpha, \beta]|^2 \right)$$

as this generalizes both eq.(1) for the non-ideal interferometer $v \neq 1$, and the QM case $P_D^{QM} = \frac{1}{2}(1 - v)$.

As shown in Methods, this is valid only if i commutes with α and β , so this is a very special case of quaternions, since $ij = k, jk = i, ki = j$ in eq.(8). Under which condition $[i, \alpha] = 0 = [i, \beta]$ so that the above equation for P_D holds?

2. Since the signal P_D is small, the Authors interfere this signal in a Mach-Zehnder-like interferometer (MZI). The experimental setup uses two beamsplitters BS1/BS2 instead of the standard mirrors of a MZI. This is necessary since the input of the Sagnac is the same as the output (fig.2).

Is there an advantage of using the MZI (which introduces extra errors and losses at BS1/BS2) instead of measuring P_D directly after the Sagnac? My concern is that this modification of the standard MZI introduces extra losses in a situation where the signal P_D to be measured is already small to start with.

An alternative setup would be the one shown in the next figure, in which the input and output of the Sagnac are displaced, and thus easier to separate. In this case BS1/BS2 are replaced by standard mirrors, hence avoiding the losses at BS1/BS2.

3. We can rewrite eq.(3) as

$$V^2 + v^2\Gamma^2 = 1$$

Thus the relevant factor is the product $v\Gamma$: a loss in this factor can come either from the visibility v or from Γ . Alternatively, a decrease in Γ can be compensated by an increase in v , for the same measured visibility V of the MZI. How can we distinguish between these two cases without making further assumptions about v (see next comment)?

4. pg.5, first paragraph: "if we perform two different measurements, each with different phases in the interferometer (two different values of Γ), we can altogether avoid needing to know v the visibility of the Sagnac interferometer."

This assumes that the visibility v is the same for both phases. Since the MZI visibilities V_0 and V_{NIM} are (experimentally) different for different phases, why this is not true also for the Sagnac visibilities v_0, v_{NIM} ?

5. pg.6. The NIM has a "negative phase shift $-\pi$ ". I think this is a misnomer, since a phase of π is the same as a "negative" phase $-\pi$. One cannot distinguish a "negative" phase shift α from a "positive" phase shift $2\pi - \alpha$.

In conclusion I recommend publication once the Authors have answered the above questions.

Reviewer #2 (Remarks to the Author):

One relevant question in the foundations of quantum mechanics, therefore in our understanding of nature, is the reason for the use of complex Hilbert spaces in the description of quantum phenomena. Two other mathematically consistent alternatives are available: real space and quaternionic spaces. We know that the first alternative is ruled out by its lack to describe the large variety of quantum effects. Much less is known experimentally about the second alternative.

The paper reports on a novel experiment, which sets a stronger bound than previous experiments on quaternionic quantum theories. The fundamental idea is that while complex phases commute, quaternionic phases do not commute in general. Experimentally this is tested by sending photons through a Sagnac interferometer, with (essentially) two phase shifters. By inverting the order of the phase shifters, the authors check the commutativity of phases, and place a bound, as reported in the text.

The paper is clear in its exposition, detailed, well written, and of interest to the community referring to Nature Communications. I recommend it for publication.

I have a remark regarding the first paragraph of the introduction. They write "In many post-quantum theories although the mathematical formalism changes, the experimental predictions remain the same. However, some make novel experimental predictions, and experimental tests of such post-quantum theories have only recently begun". Here it is odd that the authors do not refer to the rather large literature on experimental tests of collapse models, which is attracting a significant attention.

In general, the introductory paragraph (rather important for a paper in a journal of the Nature group) is misleading. The proposals they quote as "maintaining elements of classicality" actually all "possess inherently quantum features, such as superposition and "non-locality", so the division they make does not make sense. Why not stating plainly that there are alternatives to standard quantum mechanics. Some are experimentally equivalent, some not. Quaternionic quantum mechanics falls in this second class. This would give the reader a much clearer view of the context in which this paper finds its natural location.

Typos.

1. Page 9 of the manuscript, 5th line of the last paragraph. It should be: "IT was shown that quaternionic effects ..."
2. Page 11 of the manuscript, line before Eq. (9). It should be: "we can write THE state in ..."

Reviewer #3 (Remarks to the Author):

Summary:

The paper deals with the following:

- They look at a particular consequence of quaternionic quantum theory that marks a departure from standard quantum theory.
- i.e., that the non-commutativity of quaternionic phases implies a lower visibility in the interference pattern their experiment is designed to generate.
- They design and conduct an experiment to test this claim, and obtain a null result to a precision an order of magnitude better than previous experiments.
- They point out that their experiment is the first to use relativistic particles (i.e., photons). The

only other experiment used comparably slow neutrons, where subsequent analysis showed that quaternionic effects, if any, would be impractically difficult to observe.

Overall comments:

It is a well organized paper with a clear purpose. It is likely of interest within the quantum physics community, as it describes a practical test with the potential to falsify elements of standard quantum mechanics, even though a null result was obtained.

There does not seem to be sufficient discussion as to why the properties of optical media should produce anything other than a complex-valued response in the photon. Can one solve a quaternionic Schrodinger equation to derive a quaternionic optical susceptibility tensor for a given optical medium? This should be explained more clearly. Without such motivation, there is no reason to expect anything other than a null result, which would greatly diminish the overall interest (though not the validity) of the paper.

As is, the paper is not of sufficiently broad interest for publication in Nature Communications, though it would be well-suited in a more subject-specific journal of comparable impact. With significant edits to explicate the motivation to readers unfamiliar with quaternion-based quantum theory, the paper would be acceptable.

Below we respond point-by-point to all of the Reviewers' comments.

Reviewer 1

Speaking about the impact of our work, Reviewer 1 said:

"The article is well-written and the subject is of interest for quantum foundations. The results are timely and important for future developments of post-quantum theories, as they exclude a class of theories incompatible with experiments."

This Reviewer then went on to provide a detailed report, and raise some very good questions regarding our assumptions and data analysis. We respond in detail to these questions below. Given that Reviewer 1 said *"in conclusion I recommend publication once the Authors have answered the above questions"*, we think that our work is now suitable for *Nature Communications*.

COMMENT 1a: *"I suggest to rewrite eq.(2) as*

$$P_D = \frac{1}{2} \left(1 - v + \frac{v}{2} |[\alpha, \beta]|^2 \right)$$

as this generalizes both eq.(1) for the non-ideal interferometer $v \neq 1$, and the QM case $P_D^{QM} = \frac{1}{2}(1 - v)$."

RESPONSE 1a: We have re-written eq. (2) as the reviewer suggested.

COMMENT 1b: *"As shown in Methods, this is valid only if i commutes with α and β , so this is a very special case of quaternions, since $ij = k, jk = i, ki = j$ in eq.(8). Under which condition $[i, \alpha] = 0 = [i, \beta]$ so that the above equation for P_D holds?"*

RESPONSE 1b: In quaternionic quantum mechanics this condition does not hold in general. This is why in the Methods section, we give the general equation (Eq. 14). The general equation differs from Eq. 2 in that the commutator $[\alpha, \beta] = \alpha\beta - \beta\alpha$ is replaced with a very similar quantity $i\alpha\beta - \beta\alpha i$. This new quantity is zero if α, β , and i all mutually commute, and if $[i, \alpha] = 0 = [i, \beta]$ it simplifies to the commutator $[\alpha, \beta]$. Thus whenever $P_D \neq 0$ there is some form of non-commutativity. However, we cannot attribute it directly to a failure of α and β to commute, but rather to a failure of the three phases α, β , and i mutually commute.

Since this would still be a failure of phases to commute because of quaternionic quantum mechanics this is just as interesting as characterizing the commutativity of only α and β . However, we feel that adding too much of this discussion to main text unnecessarily complicates the manuscript. In our original manuscript this discussion was already in the Methods section. However, we now explicitly refer to the general equation in the Methods section after this equation, and we have expanded the discussion in the Methods section.

In particular, after Eq. 2 we say: “Eq. 2 is in fact a special case of the general equation presented in the methods section (Eq. 14), assuming that two phases commute with the reflection phase of the beamsplitter.” Moreover, in the Methods section we added the following text:

“Notice that P_D defined here differs slightly from the Eq. 2 of the main text, in that $|\alpha\beta - \beta\alpha i|$ replaces $|\alpha, \beta|$. However, a non-zero value of this new quantity would also signify a deviation of QQM from CQM, and is, thus, also interesting to study.”

COMMENT 2: “Since the signal P_D is small, the Authors interfere this signal in a Mach-Zehnder-like interferometer (MZI). The experimental setup uses two beamsplitters BS1/BS2 instead of the standard mirrors of a MZI. This is necessary since the input of the Sagnac is the same as the output (fig.2).

Is there an advantage of using the MZI (which introduces extra errors and losses at BS1/BS2) instead of measuring P_D directly after the Sagnac? My concern is that this modification of the standard MZI introduces extra losses in a situation where the signal P_D to be measured is already small to start with.

An alternative setup would be the one shown in the next figure, in which the input and output of the Sagnac are displaced, and thus easier to separate. In this case BS1/BS2 are replaced by standard mirrors, hence avoiding the losses at BS1/BS2.”

RESPONSE 2: It is true that that adding the Mach-Zehnder interferometer in our configuration introduces extra losses: it will decrease our count rate by factor of 4 (since

each photon must be transmitted through two 50/50 beamsplitters). Therefore, if we were to send N photons into our apparatus at most $N/4$ of them would be transmitted. Thus, our statistical uncertainty scales as $\sqrt{4/N}$, rather than $\sqrt{1/N}$. In other words, our statistical error is larger by a factor 2. However, this is a constant factor, which can always be compensated for by acquiring more data.

Additionally, when we interfere the two output modes of the Sagnac interferometer the signal. In particular, we measure an oscillating signal with an amplitude of $\sqrt{(1 - P_D)P_D}$. This signal is increased by a factor of $\sqrt{\frac{1-P_D}{P_D}}$, compared to a direct measurement of P_D . Since P_D is less than 0.005 in our experiment the effect of this amplification is very significant. To clarify this, we have added the following sentence to our manuscript “This visibility is amplified, with respect to a direct measurement of P_D , by a factor of $\sqrt{\frac{1-P_D}{P_D}}$.”

The Reviewer then suggests an alternative setup, which also interferes the output modes. In this proposed setup the Sagnac interferometer is not a common-path interferometer, as in our experiment. Therefore, the Reviewer’s set-up does not introduce the additional losses that ours does and would decrease our statistical error by a factor of 2, while providing the same amplification. While we have not studied this proposed setup experimentally, we are concerned that it may introduce additional systematic error since the two beams pass through different positions of the optics in the Sagnac interferometer. Thus any spatial imperfections could potentially play a larger role and potentially decrease the visibility of the Sagnac interferometer.

COMMENT 3: “We can rewrite eq. (3) as

$$V^2 + v^2 \Gamma^2 = 1$$

Thus the relevant factor is the product $v\Gamma$: a loss in this factor can come either from the visibility v or from Γ . Alternatively, a decrease in Γ can be compensated by an increase in, for the same measured visibility V of the MZI. How can we distinguish between these two cases without making further assumptions about v (see next comment)?”

RESPONSE 3: Here, Reviewer 1 basically asks to clarify how we can distinguish between the possible quaternionic effects, which lead to a decreased value of Γ , and a change in the visibility of the Sagnac interferometer v . The Reviewer is correct that without making

assumptions about v , we cannot distinguish between these two effects. However, we took great pains to ensure that the visibility of Sagnac interferometer was independent of the liquid-crystal phase shift. For a given run, this can be seen by comparing the signals shown in Fig. 3a and 3b. To make this even clearer we have additional data to the Supplementary Information, showing our detailed statistical analysis. In this data (the newly-added Figure 8 of the Supplementary Information), we see that the liquid-crystal phase shifter has no observable effect on the interferometer.

On the other hand, inserting the meta-material did change the visibility of the Sagnac interferometer. This leads into the Reviewer's next comment, which we will discuss more in the next point. However, in brief, this is we the left the meta-material in place and only varied the liquid crystal phase shifter to acquire our final data.

COMMENT 4: *"pg.5, first paragraph: "if we perform two different measurements, each with different phases in the interferometer (two different values of Γ), we can altogether avoid needing to know v the visibility of the Sagnac interferometer."*

This assumes that the visibility v is the same for both phases. Since the MZI visibilities V_0 and V_{NIM} are (experimentally) different for different phases, Why this is not true also for the Sagnac visibilities v_0, v_{NIM} ?"

RESPONSE 4: This is related to Comment 3. The visibilities of the Sagnac interferometer with and without the meta-material (v_0 and v_{NIM} , respectively) are not equal, and we do not need to assume that they are equal. However, we do assume that the visibility of the Sagnac interferometer with only the meta-material inserted v_{NIM} is equal to the visibility of the Sagnac interferometer with both the meta-material and the liquid-crystal inserted v_{NIM+LC} . As we stated above, we already performed a very detailed measurement showing that the visibility of the Sagnac interferometer is independent of the liquid-crystal phase-shifter, and we have added additional data to the paper showing this. In this case, any change to the visibility of the Mach-Zehnder interferometer with and without the liquid-crystal phase shifter (i.e. a difference between V_{NIM+LC} and V_{NIM}) must come from Γ , which is related to the commutativity of the phases.

We have added the following sentences in the main text in order to clarify this point. First, on page 5, where the Reviewer raises the point, we now state “To be clear, to use this fact one must ensure that the visibility of the Sagnac interferometer is unchanged by the addition of the phase.” Then, later, we say: “This visibility increase occurs because inserting the NIM slightly degrades or shifts the spatial modes inside the Sagnac interferometer. We believe that this increase in visibility is rather a systematic error (i.e. a decrease in the visibility of the Sagnac interferometer), and not the quaternionic effect that we are interested in.”

COMMENT 5: *“pg. 6. The NIM has a “negative phase shift $-\pi$ ”. I think this is a misnomer, since a phase of π is the same as a “negative” phase $-\pi$. One cannot distinguish a “negative” phase shift α from a “positive” phase $2\pi - \alpha$.”*

RESPONSE 5: We agree with Reviewer 1 that one cannot distinguish a “negative” phase shift α from a “positive” phase $2\pi - \alpha$.” However, the term “negative phase shift” has a physical meaning, and is a commonly accepted term in the meta-materials community. As we explain in the Supplementary Information, the Poynting vector and wave vector are anti-parallel for light propagation inside the NIM. By comparing our refractive index simulation and our broadband measurement of the refractive index (see Figure 10), we are able us to track the phase as a function of wavelength and resolve this ambiguity. Additionally, we show simulations for the time evolution of the phase front for the normalized electric field going from the positive refractive index (air) to negative refractive index (NIM). This shows the gradual negative phase accumulation and backward wave propagation behavior. The phase is defined negative when the Poynting vector is antiparallel to the wave vector.

We have added the following sentence in the main text in order to clarify this point: “Note that the phase is negative, in the sense that the Poynting vector points opposite to the propagation vector [46], see Supplementary information for more details.” We also added reference 46, which supports this claim.

Reviewer 2

Reviewer 2 explicitly recommended our paper for publication, saying:

“the paper is clear in its exposition, detailed, well written, and of interest to the community referring to Nature Communications. I recommend it for publication.”

Reviewer 2’s only other remarks regarding our work were related to missing references, the introductory paragraph, and two minor typos. Below we respond to these comments.

COMMENT 1: *“I have a remark regarding the first paragraph of the introduction. They write “In many post-quantum theories although the mathematical formalism changes, the experimental predictions remain the same. However, some make novel experimental predictions, and experimental tests of such post-quantum theories have only recently begun”. Here it is odd that the authors do not refer to the rather large literature on experimental tests of collapse models, which is attracting a significant attention”*

RESPONSE 1: We thank the Reviewer for pointing this out, and we are chagrined for missing such important references. We have now added references to this representative work (references 17-19 in the resubmitted manuscript).

COMMENT 2: *“In general, the introductory paragraph (rather important for a paper in a journal of the Nature group) is misleading. The proposals they quote as “maintaining elements of classicality” actually all “possess inherently quantum features, such as superposition and “non-locality”, so the division they make does not make sense. Why not stating plainly that there are alternatives to standard quantum mechanics. Some are experimentally equivalent, some not. Quaternionic quantum mechanics falls in this second class. This would give the reader a much clearer view of the context in which this paper finds its natural location.”*

RESPONSE 2: In our original manuscript we may not have been clear enough about our intended distinction between post-quantum theories and those “maintaining elements of classicality”. We do feel that this distinction is important, although we may have worded our original manuscript too strongly and not explained ourselves well enough. To be clear, the theories we referred to as “maintaining elements of classicality” are not fully classical theories, rather they typically attempt to explain away one “interpretationally troubling” aspect of quantum mechanics. For example, hidden-variable theories attempt to do away with non-locality. Post-quantum theories, on the other hand, are interested in finding new predications, which go beyond quantum mechanics. They do not attempt to resolve these “problems”.

To make this clearer we completely reworded our introductory paragraph. We no longer refer to these theories as “classical-like”, but we say they “attempt to save one or another feature of classical physics.” We further define them to make the distinction clearer, saying:

“In the past decades, quantum mechanics has been challenged mainly from a conceptual and interpretational perspective, via a variety of alternative theories that mainly attempt to save one or another feature of classical physics; i.e. determinism, localizability, or macroscopic realism. These alternatives include various hidden-variable models, non-linear modifications of quantum dynamics, and spontaneous localization models, amongst others.”

We also now explicitly say about post-quantum theories:

“On the other hand, in the operational approach to a physical theory, quantum mechanics is just one particular theory in a vast sea of generalized probabilistic theories.”

Then, later:

“These generalized theories are known as *post-quantum theories*. Many of them share features typically thought as quantum, such as non-locality, no-signaling, or no-cloning.

We believe that this new introductory paragraph makes these issues much clearer, and gives more credit to alternative viewpoints. We hope that the Reviewer agrees.

COMMENT 3:

Typos.

- 1. Page 9 of the manuscript, 5th line of the last paragraph. It should be: “IT was shown that quaternionic effects ...”*
- 2. Page 11 of the manuscript, line before Eq. (9). It should be: “we can write THE state in ...”*

RESPONSE 3: We have fixed these typos.

Reviewer 3

Reviewer 3’s overall comments about our work were generally positive:

“It is a well organized paper with a clear purpose. It is likely of interest within the quantum physics community, as it describes a practical test with the potential to falsify elements of standard quantum mechanics, even though a null result was obtained.”

The Reviewer then had comments related to the motivation of our experiment, and the introductory paragraph. As such, we have revised our introduction (we also did this in response to Reviewer 2’s comments). Our new introduction makes our motivations clear to

non-experts, and thus we think that our paper suitable for publication in *Nature Communications*.

COMMENT 1: *“There does not seem to be sufficient discussion as to why the properties of optical media should produce anything other than a complex-valued response in the photon. Can one solve a quaternionic Schrodinger equation to derive a quaternionic optical susceptibility tensor for a given optical medium? This should be explained more clearly. Without such motivation, there is no reason to expect anything other than a null result, which would greatly diminish the overall interest (though not the validity) of the paper.”*

RESPONSE 1: We agree with Reviewer 3 that if one wishes to predict any deviation from complex quantum mechanics, one has to invoke a model to characterize interactions between the photon and the optical medium. Adler has shown that for non-relativistic particles all quaternionic effects are exponentially damped, and, therefore, not experimentally accessible (as in the previous neutron interference experiments). However, there is still an on-going debate as to whether this applies to relativistic particles. This is because there is no quaternionic quantum field theory, so it is not currently possible to make a specific prediction for a photonic system. However, exciting preliminary calculations show that quaternionic effects may persist for relativistic particles [31]. This makes photonic experiments excellent candidates to test for quaternionic effects. This was a strong motivation for our experiment, and we hope that Reviewer 3 agrees (as Reviewers 1 and 2 already did).

We clearly state this in the manuscript: “However, there is strong theoretical evidence that quaternionic effects will persist for relativistic particles, such as single photons. In fact, we recently showed that for relativistic Klein-Gordon scattering quaternionic effects do indeed persist [31].”

COMMENT 2: *“As is, the paper is not of sufficiently broad interest for publication in Nature Communications, though it would be well-suited in a more subject-specific journal of comparable impact. With significant edits to explicate the motivation to readers unfamiliar with quaternion-based quantum theory, the paper would be acceptable”*

RESPONSE 2: We have significantly expanded the introductory paragraph in order address this comment. We think that our paper is of broad interest and suitable for publication in *Nature Communications* for three main reasons. First, in our experiment we test a very fundamental question: Is quantum mechanics based on complex numbers or something

more general? This is question that already interested von Neumann, who studied a quantum mechanics based on real numbers. Furthermore, every physicist learns that quantum mechanics requires complex numbers (rather than simple real numbers); therefore, all physicists will relate to this motivation. Secondly, our experiment is the first experiment to test for quaternionic effects in the relativistic regime, wherein very recent theoretical work has shown that quaternionic effects may manifest themselves. Finally, our work fits into the more general theme of testing post-quantum theories. This perspective provides a unique and modern viewpoint. Thus our work tests a decades-old issue, from a modern point of view in a completely new regime. We certainly feel that this will be of interest to the broader community, and we hope that our new introduction makes this motivation clear.

Conclusion: We trust that we have responded to the all of the concerns of the Reviewers, and we think these changes have made the presentation of our work even more accessible to the broader audience.

Reviewer #1 (Remarks to the Author):

The Authors answered all my comments and therefore I recommend publication.

Reviewer #2 (Remarks to the Author):

In relation to my first report and the author's reply, I still disagree with the issue of "classicality". It does not make sense. Take standard quantum mechanics, in the Copenhagen interpretation as given by Bohr. He repeatedly stated that measuring devices should be described by classical physics. Therefore this also should be accounted as a theory that saves "one or another feature of classical physics", and in particular "macroscopic realism." Any theory has to have some "elements of classicality" for the very simple reason that it must include classical mechanics in the limiting case.

Reading further on, they speak of "localizability" among the "elements of classicality", and quote the GRW model. Again, this is misleading and is some sense wrong. In the GRW model, quantum systems are not localized. Only macro-objects are. Then as for "elements of classicality", also localizability is an empty word, if not better specified (Localizability of what? Under which conditions? And so on).

Next, among the alternatives they cite "non-linear modifications of quantum dynamics" and quote a paper of Weinberg (ref 16). The abstract of the cited paper states "This paper presents a general framework for introducing nonlinear corrections into ordinary quantum mechanics, that can serve as a guide to experiments that would be sensitive to such corrections." It is not concerned with "classicality", but with extensions of quantum theory, and their experimental testability. Therefore it should better fall in the class of what the authors of the manuscript refer to as "generalized theories". Incidentally, in the same years as Weinberg's paper came out, it was proven that these nonlinear theories are physically inconsistent (N. Gisin, *Hel. Phys. Acta* 62, 363 (1989) and *Phys. Lett. A* 143, 1 (1990); N. Gisin and M. Rigo, *Journ. Phys. A* 28, 7375 (1995); J. Polcinski, *Phys. Rev. Lett.* 66, 397 (1991)). This should also be quoted. The same kind of criticism holds for ref. 15.

Again incidentally, in quoting hidden variable theories, the only theoretical paper quoted, among several experimental ones, is that of Bohm (1952), except for general papers by Einstein and Bell. Much more happened from 1952 to today, on the theoretical side. At least, the authors could quote the more recent literature, like the book: Dürr, Detlef, Teufel, Stefan "Bohmian Mechanics" (Springer 2009).

Next, in their "response 1" to my previous report, they added references 17-19 to the collapse models literature. This is an unjustified choice, rather biased given that all three quoted papers refer to the same group, whose main research activity is not on models of spontaneous wave function collapse. Just as an example, there is a rather well known review in the literature: A. Bassi, K. Lochan, S. Satin, T.P. Singh and H. Ulbricht: "Models of Wave-function Collapse, Underlying Theories, and Experimental Tests", *Rev. Mod. Phys.* 85, 471 (2013), or the paper: A. Vinante, M. Bahrani, A. Bassi, O. Usenko, G. Wijts, T.H. Oosterkamp, "Upper bounds on spontaneous wave-function collapse models using millikelvin-cooled nanocantilevers", *Phys. Rev. Lett.* 116, 090402 (2016), which gives more significant bounds than those relative to refs 18 and 19.

Reading further on, the authors write "These generalized theories are known as post-quantum theories. Many of them share features typically thought as "quantum", such as "non-locality", "no-signaling," or "no-cloning." According to which criterion is "no-signaling" supposed to be a "quantum" feature? It is a condition required by relativity, not by quantum mechanics. Same for "non-locality": Newtonian mechanics, without relativity, is non-local. There is nothing "quantum" in

the concept of "non-locality".

I wish to add one more thing, relative not to the new version of the manuscript, but to the reply. They write "For example, hidden-variable theories attempt to do away with non-locality". This is simply false. Actually, it is the opposite. The first and best example of a hidden variable theory is Bohmian Mechanics. This is a nonlocal theory. It is because of its nonlocality, that quantum nonlocality was discovered (Bell's work). Just the opposite of doing away with non-locality.

To conclude, I stress again what I wrote in my first report. Especially for a paper on a Journal of the Nature group, the introduction is the business card of the entire work. It should be appropriately written. This is not. I advise the authors to cut the story short, by simply listing alternatives to quantum mechanics among which there are quaternionic theories, which they test. And actually I do not understand why they make the story so complicated (and confused). Quaternionic theories are interesting and are one of the attempts at generalizing, or modifying quantum theory. Each attempt has its own history and motivations. Trying to classify such motivations is a dangerous task, and is not needed for the present work.

Reviewer #3 (Remarks to the Author):

In the revised manuscript, the Authors have successfully explained the broad appeal of their research. Though I still maintain that being able to predict a quaternionic response in an optical material would substantially improve the paper, the new experimental approach described here is sufficiently important to future research to warrant publication in Nature Communications.

Here we respond point-by-point to Reviewer 2's comments. All of the Reviewer's comments dealt only with the first paragraph. Since there is significant overlap between the various comments, we describe our response to each of the Reviewer's points, and paste our revised introductory paragraph only once at the end.

- 1) *"In relation to my first report and the author's reply, I still disagree with the issue of "classicality". It does not make sense. Take standard quantum mechanics, in the Copenhagen interpretation as given by Bohr. He repeatedly stated that measuring devices should be described by classical physics. Therefore this also should be accounted as a theory that saves "one or another feature of classical physics", and in particular "macroscopic realism." Any theory has to have some "elements of classicality" for the very simple reason that it must include classical mechanics in the limiting case.*

The Reviewer makes a valid point regarding the classicality in the Copenhagen interpretation. In any case, as suggested by the Reviewer we have removed all discussion of the role of classicality in different interpretations.

- 2) *"Reading further on, they speak of "localizability" among the "elements of classicality", and quote the GRW model. Again, this is misleading and is some sense wrong. In the GRW model, quantum systems are not localized. Only macro-objects are. Then as for "elements of classicality", also localizability is an empty word, if not better specified (Localizability of what? Under which conditions? And so on)."*

We may have misrepresented this point in an attempt at brevity. The GRW model indeed allows microscopic quantum systems to be non-local. As the Reviewer suggested we have removed discussion of localizability.

- 3) *"Next, among the alternatives they cite "non-linear modifications of quantum dynamics" and quote a paper of Weinberg (ref 16). The abstract of the cited paper states "This paper presents a general framework for introducing nonlinear corrections into ordinary quantum mechanics, that can serve as a guide to experiments that would be sensitive to such corrections." It is not concerned with "classicality", but with extensions of quantum theory, and their experimental testability. Therefore it should better fall in the class of what the authors of the manuscript refer to as "generalized theories". Incidentally, in the same years as Weinberg's paper came out, it was proven that these nonlinear theories are physically inconsistent (N. Gisin, Hel. Phys. Acta 62, 363 (1989) and Phys. Lett. A 143, 1 (1990); N. Gisin and M. Rigo, Journ. Phys. A 28, 7375 (1995); J. Polcinski, Phys. Rev. Lett. 66, 397 (1991)). This should also be quoted. The same kind of criticism holds for ref. 15.*

Given that we have removed the discussion of "classicality" we think that these references (15 and 16 in our previous manuscript) now apply to "non-linear modifications of quantum dynamics". Thus we have left them where they were, and we added the additional references suggested by the reviewer.

- 4) *Again incidentally, in quoting hidden variable theories, the only theoretical paper quoted, among several experimental ones, is that of Bohm (1952), except for general papers by Einstein and Bell. Much more happened from 1952 to today, on the theoretical side. At least, the authors could quote the more recent literature, like the book: Dürr, Detlef, Teufel, Stefan "Bohmian Mechanics" (Springer 2009).*

We have added this reference.

- 5) *Next, in their "response 1" to my previous report, they added references 17-19 to the collapse models literature. This is an unjustified choice, rather biased given that all three quoted papers refer to the same group, whose main research activity is not on models of spontaneous wave function collapse. Just as an example, there is a rather well known review in the literature: A. Bassi, K. Lochan, S. Satin, T.P. Singh and H. Ulbricht: "Models of Wave-function Collapse, Underlying Theories, and Experimental Tests", Rev. Mod. Phys. 85, 471 (2013), or the paper: A. Vinante, M. Bahrami, A. Bassi, O. Usenko, G. Wijts, T.H. Oosterkamp, "Upper bounds on spontaneous wave-function collapse models using millikelvin-cooled nanocantilevers", Phys. Rev. Lett. 116, 090402 (2016), which gives more significant bounds than those relative to refs 18 and 19.*

We apologize for this oversight. We have added the suggested references.

- 6) *Reading further on, the authors write "These generalized theories are known as post-quantum theories. Many of them share features typically thought as "quantum", such as "non-locality", "no-signaling," or "no-cloning." According to which criterion is "no-signaling" supposed to be a "quantum" feature? It is a condition required by relativity, not by quantum mechanics. Same for "non-locality": Newtonian mechanics, without relativity, is non-local. There is nothing "quantum" in the concept of "non-locality".*

By no-signalling as a quantum feature we meant a general "no-signalling theorem" which can be derived within standard quantum theory, wherein a reduced density matrix describing system B cannot depend on transformations or dynamics applied on a distant system A. We did not intend to go into interpretational issues related to non-locality, "spooky-action at a distance", or collapse of the wavefunction at this point. Thus, we have removed the discussion both of non-locality and no-signalling.

- 7) *I wish to add one more thing, relative not to the new version of the manuscript, but to the reply. They write "For example, hidden-variable theories attempt to do away with non-locality". This is simply false. Actually, it is the opposite. The first and best example of a hidden variable theory is Bohmian Mechanics. This is a nonlocal theory. It is because of its nonlocality, that quantum nonlocality was discovered (Bell's work). Just the opposite of doing away with non-locality.*

The Reviewer is correct; we misspoke. We meant that hidden variable theories attempt to ascribe an objective reality. However, since this was in our previous response (not in the manuscript) we have made no changes to the manuscript.

- 8) *To conclude, I stress again what I wrote in my first report. Especially for a paper on a Journal of the Nature group, the introduction is the business card of the entire work. It should be appropriately written. This is not. I advise the authors to cut the story short, by simply listing alternatives to quantum mechanics among which there are quaternionic theories, which they test. And actually I do not understand why they make the story so complicated (and confused). Quaternionic theories are interesting and are one of the attempts at generalizing, or modifying quantum theory. Each attempt has its own history and motivations. Trying to classify such motivations is a dangerous task, and is not needed for the present work.*

We have completely followed this advice. We now simply list alternatives to standard quantum theory, and have removed any attempts at to differentiate general-probabilistic theories from other alternatives. In our revised introduction they are all treated on the same footing.

Revised first paragraph:

“Quantum mechanics is an extremely well-established scientific theory. Although it has been contested since its inception, beginning with the famous “Bohr-Einstein debates¹,” quantum mechanics has stood its ground against competing theories and experimental tests for almost 100 years. Especially in the past decades, quantum mechanics has been challenged by a variety of alternative theories, including various hidden-variable models²⁻¹¹, non-linear modifications of quantum dynamics¹²⁻¹⁷, spontaneous localization models¹⁸⁻²³, and generalized probabilistic theories²⁴⁻³⁰. In generalized probabilistic theories, sometimes also called *post-quantum theories*, quantum mechanics is just one particular theory in a vast sea of possibilities. One important class within this sea are the so-called hyper-complex theories²⁶⁻³¹. They differ from standard quantum theory in the nature of superposition coefficients (probability amplitudes). Whether nature prefers a quantum theory based on real, complex, quaternionic, or general hyper-complex amplitudes is an experimental issue. Excitingly, some predictions of hyper-complex are experimentally testable, since basing the superposition principle on hyper-complex probability amplitudes leads to a version of quantum mechanics wherein simple phases are not guaranteed to commute^{32,33}. Although this prediction has experimentally been studied in the past with massive, non-relativistic particles³³, in that regime the quaternionic amplitudes are known to be exponentially suppressed^{27,34}. Thus it is difficult to place bounds on genuine post-quantum effects in these experiments. However, new theoretical calculations have shown that this result does not necessarily apply to relativistic particles, such as single photons³⁵. Therefore, any discrepancies between standard complex quantum theory and its hyper-complex generalization may be experimentally accessible in the relativistic regime.”